# Continual Learning In Environments With Polynomial Mixing Times

**Matthew Riemer**[*1, 2], **Sharath Chandra Raparthy**[*2], **Ignacio Cases**[3], **Gopeshh Subbaraj**[2], **Maximilian Puelma Touzel**[2], **Irina Rish**[2]

## Abstract

The mixing time of the Markov chain induced by a policy limits performance in real-world continual learning scenarios. Yet, the effect of mixing times on learning in continual reinforcement learning (RL) remains underexplored. In this paper, we characterize problems that are of long-term interest to the development of continual RL, which we call scalable MDPs, through the lens of mixing times. In particular, we theoretically establish that scalable MDPs have mixing times that scale polynomially with the size of the problem. We go on to demonstrate that polynomial mixing times present significant difficulties for existing approaches, which suffer from myopic bias and stale bootstrapped estimates. To validate our theory, we study the empirical scaling behavior of mixing times with respect to the number of tasks and task duration for high performing policies deployed across multiple Atari games. Our analysis demonstrates both that polynomial mixing times do emerge in practice and how their existence may lead to unstable learning behavior like catastrophic forgetting in continual learning settings.

## 1 Introduction

Continual reinforcement learning (RL) [1] is an aspirational field of research confronting the difficulties of long-term, real-world applications by studying problems of increasing scale, diversity, and non-stationarity. The practical requirement for researchers to work on problems of reasonable complexity in the short-term presents a meta-challenge: choosing the right small-scale problems so that the approaches we develop scale up to the use cases of the future. Here, we address this meta-challenge by formalizing RL problems that vary in size and by analyzing the scaling behavior of popular RL algorithms. In particular, we analyze on the often ignored *mixing time* that expresses the amount of time until the agent-environment dynamics converge to some stationary behaviour.

Towards this end, we specifically make the following contributions in this work:

1. **Scalable MDPs:** We propose the formalism of *scalable MDPs* in Definition 1 to characterize an abstract class of MDPs where the MDPs within the class are differentiated based on a changing scale parameter. Understanding the influence of this scale parameter on learning can promote better understanding of the meta-challenge of extrapolating our results on small scale problems to large scale problems of the same class.

2. **Polynomial Mixing Times:** Theorem 1 establishes the key result of this paper that as any scalable MDP is scaled, its mixing time must grow polynomially as a function of the growing state space. This has major implications for regret analysis in these MDPs.

3. **Myopic Bias During Scaling:** We demonstrate in Corollaries 2, 3, and 4 that Theorem 1 implies traditional approaches to RL cannot efficiently scale to problems of large size without experiencing myopic bias in optimization that slows down learning.

---

[*]Equal Contribution
[1]IBM Research; [2]Mila, Université de Montréal; [3]Massachusetts Institute of Technology

36th Conference on Neural Information Processing Systems (NeurIPS 2022).

4. **Empirical Analysis of Mixing for Continual RL:** We back up our theory with empirical analysis of mixing time scaling in continual RL settings based on the Atari and Mujoco benchmarks. Our analysis provides insight into why agents experience significant catastrophic forgetting or other optimization instability when learning in these domains. We hope that our analysis into the fundamental driver of these issues can open the door for more successful and principled approaches to continual RL domains like these moving forward.

We refer readers to Appendix D for detailed proofs of all propositions and theorems in our paper.

## 2 Formalizing Our Continual RL Setting

Aiming to characterize a broad range of settings, we focus on the formulation of RL in continuing environments. As explained in Khetarpal et al. [1], not only are supervised learning and episodic RL special cases of RL in continuing environments, but so are their non-stationary variants (continual supervised learning and continual episodic RL), despite violating the stationarity assumptions of their root settings. Indeed, approaches that do not acknowledge the more general RL in continuing environments setting exhibit myopic bias in their optimization when faced with non-stationarity [1].

Unfortunately, the popular discounted reward setting inserts the very same kind of myopic bias in optimization that we would like to avoid [2]. As typically implemented, discounting does not correspond to the maximization of any objective function over a set of policies [3] and the policy gradient is not the gradient of any function [4]. These fundamental issues do not resolve as the discount factor approaches 1 [3] and discounting does not influence the ordering of policies, suggesting it likely has no role to play in the definition of the control problem [5]. In contrast, the average reward per step objective, which explicitly includes an average over the stationary distribution, avoids inducing myopic biases and hence is well-suited for continual RL problems [5, 1].

### 2.1 Average Reward RL in Continuing Environments

RL in continuing environments is typically formulated using a finite, discrete-time, infinite horizon Markov Decision Process (MDP) [6, 5], which is a tuple $\mathcal{M} = \langle \mathcal{S}, \mathcal{A}, T, R \rangle$, where $\mathcal{S}$ is the set of states, $\mathcal{A}$ is the set of actions, $R : \mathcal{S} \times \mathcal{A} \to [0, R^{\max}]$ is the reward function, and $T : \mathcal{S} \times \mathcal{S} \times \mathcal{A} \to [0, 1]$ is the environment transition probability function. At each time step, the learning agent perceives a state $s \in \mathcal{S}$ and takes an action $a \in \mathcal{A}$ drawn from a policy $\pi : \mathcal{S} \times \mathcal{A} \to [0, 1]$ with internal parameters $\theta \in \Theta$. The agent then receives a reward $R(s, a)$ and with probability $T(s'|s, a)$ enters next state $s'$. Markov chains may be periodic and have multiple recurrent classes, but optimality is difficult to define in such cases [7], making the following assumption necessary for analysis:

**Assumption 1** *All stationary policies are aperiodic and unichain, meaning they give rise to a Markov chain with a single recurrent class that is recurrent in the Markov chain of every policy.*[2]

Any RL problem may be modified such that Assumption 1 holds by adding an arbitrarily small positive constant $\epsilon$ to all transition probabilities in $T(s'|s, a)$ and renormalizing in which case the effect on the objective of each stationary policy is $O(\epsilon)$ [8]. An important corollary to Assumption 1 is that the *steady-state distribution* $\mu^\pi$ induced by the policy $\pi$ is independent of the initial state:

**Corollary 1** *All stationary policies $\pi$ induce a unique steady-state distribution $\mu^\pi(s) = \lim_{t \to \infty} P^\pi(s_t = s | s_0)$ that is independent of the initial state such that $\sum_{s \in \mathcal{S}} \mu^\pi(s) \sum_{a \in \mathcal{A}} \pi(a|s) T(s'|s, a) = \mu^\pi(s') \ \forall s' \in \mathcal{S}$.*

Corollary 1 implies that the long-term rewards of any $\pi$ will be independent of the current state. As such, the average reward per step objective $\rho(\pi)$ can be defined independently of its starting state [5]:

$$\rho(\pi) := \lim_{h \to \infty} \frac{1}{h} \sum_{t=1}^{h} \mathbb{E}_\pi \left[ R(s_t, a_t) \right] = \lim_{t \to \infty} \mathbb{E}_\pi \left[ R(s_t, a_t) \right]$$

$$= \sum_{s \in \mathcal{S}} \mu^\pi(s) \sum_{a \in \mathcal{A}} \pi(a|s) R(s, a) . \tag{1}$$

Computing the average reward with the last expression is limited by the amount of time the Markov chain induced by the policy $T^\pi(s'|s) = \sum_{a \in \mathcal{A}} \pi(a|s) T(s'|s, a)$ needs to be run for before reaching

---

[2]This corresponds to what is called an ergodicity assumption for all stationary policies in [5].

the steady-state distribution $\mu^\pi(s)$. This amount of time is referred to in the literature as the mixing time of the induced Markov chain. We denote $t^\pi_{\text{mix}}(\epsilon)$ as the $\epsilon$-*mixing time* of the chain induced by $\pi$:

$$t^\pi_{\text{mix}}(\epsilon) := \min\left\{ h \,\Big|\, \max_{s_0 \in \mathcal{S}} d_{\text{TV}}\big(P^\pi(s_h = \cdot|s_0), \mu^\pi(\cdot)\big) \le \epsilon \right\}$$

where $d_{\text{TV}}$ is the total variation distance between the two distributions. The so-called *conventional mixing time* is defined as $t^\pi_{\text{mix}} \equiv t^\pi_{\text{mix}}(1/4)$. The conventional mixing time only gives insight about distributional mismatch with respect to the steady-state distribution, which led [9] to introduce the notion of a mismatch with respect to the reward rate. The $\epsilon$-*return mixing time* is a measure of the time it takes to formulate an accurate estimate of the true reward rate. More formally, if we denote the $h$-step average undiscounted return starting from state $s$ as $\rho(\pi, s, h)$, then we define the $\epsilon$-*return mixing time* as:

$$t^\pi_{\text{ret}}(\epsilon) := \min\left\{ h \,\Big|\, |\rho(\pi, s_0, h') - \rho(\pi)| \le \epsilon, \;\; \forall s_0 \in \mathcal{S} \text{ and } \forall h' \ge h \right\} \tag{2}$$

We will come back to this definition when we present our experiments. As emphasized in Levin and Peres [10], however, one should not get bogged down in the use-case specific definitions of mixing time (we prove their equivalent scaling behaviour in fact, *c.f.* Proposition 1 ). Rather, it is their shared property of being determined by both a policy $\pi$ and environment $\mathcal{M}$, and not simply a property of the environment itself, that is important. Thus the environment's contribution can be assessed only through mixing times obtained from extreme policies such as the optimal policy, $t^{\pi^*}_{\text{mix}}$.

An alternative approach for quantifying mixing is by considering the structure of the transition matrix $T^\pi(s'|s)$ induced by $\pi$. It is well known that the mixing properties of a Markov chain are governed by the spectral gap derived from the eigenvalues of the matrix $T^\pi(s'|s)$. Unfortunately, it is difficult to reason about the spectral gap of a class of MDPs directly. Towards this end, one useful interpretation of a Markov chain is as a random walk over a graph $\mathcal{G}(\pi, T)$ with vertex set $\mathcal{S}$ and edge set $\{(s, s')\}$, for all $s$ and $s'$ satisfying $T^\pi(s'|s) + T^\pi(s|s') > 0$. The *diameter* $D^\pi$ of the Markov chain induced by $\pi$ is thus the diameter or maximal graph distance of $\mathcal{G}(\pi, T)$. It is defined using the *hitting time* $t^\pi_{\text{hit}}(s_1|s_0)$, the first time step in which $s_1$ is reached following the Markov chain $T^\pi(s'|s)$ from $s_0$,

$$D^\pi := \max_{s_0, s_1 \in \mathcal{S}} \mathbb{E}_\pi\big[t^\pi_{\text{hit}}(s_1|s_0)\big] \;\; ; \;\; D^* := \min_\pi D^\pi \;. \tag{3}$$

$D^*$ then denotes the *minimum diameter* achieved by any policy. All MDPs that follow Assumption 1 have finite diameter $D^\pi$ for all policies [11]. Levin and Peres [10] (eq. 7.4) established the relationship, $2t^\pi_{\text{mix}} \ge D^\pi$, so that the diameter provides a lower bound for the mixing time and thus serves as a very relevant quantity when conducting mixing time analyses.

## 2.2 The Role Of Tasks And Non-stationarity

The preceding single-task formulation of RL in continuing environments can, in fact, be used to formulate continual RL, which typically makes notions of multiple tasks and non-stationarity explicit. This is indeed a useful construct because arbitrary non-stationarity precludes any consistent signal to learn from, and thus further assumptions about the environment structure must be made to make progress [1]. In this paper, we consider tasks as sub-regions of the total MDP (which alternatively can be thought of as independent MDPs connected together) and we assume the transition dynamics between and within tasks are both stationary. As highlighted in [1], this setting is quite general as it is capable of modeling many continual RL problems. For example, it easily extends to partially observable variants: if the task component is not directly observed, the problem appears non-stationary from the perspective of an agent that learns from observations of only the within-task state. That said, the continual learning literature demonstrates that optimization issues are generally experienced whether task labels are observed or not [12]. Our analysis of mixing times in this paper is an attempt to understand these difficulties even when the total MDP is fully observable and stationary.

To help the reader picture $t^\pi_{\text{mix}}$ in continual RL, we offer the example depicted in Figure 1. Here, the agent's state space $\mathcal{S}$ is decomposed into a task component, $z \in \mathcal{Z}$ and a within-task component, $x \in \mathcal{X}_z$. For a policy, $\pi$, the pair of Markov chains induced by marginalizing over $x$ and over $z$ have diameters $D^\pi_z$ and $D^\pi_x$, respectively. Using the result connecting mixing times and diameters here, we can establish a universal rule for problems of this type: $2t^\pi_{\text{mix}} \ge D^\pi \ge D^\pi_x \ge D^\pi_z$, where $D^\pi$ is the diameter over the entire state space. This rule highlights the intimate connection between mixing

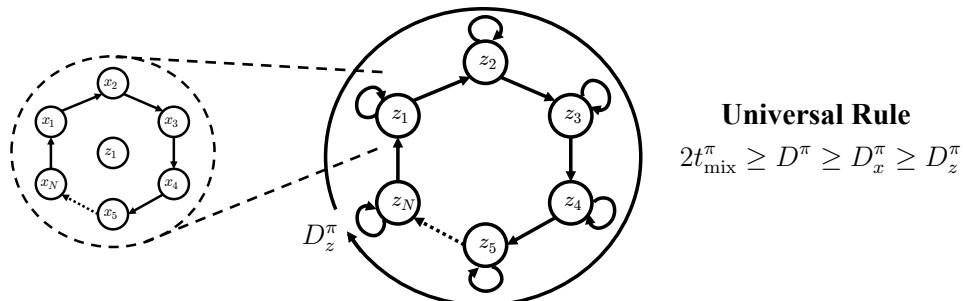

**Universal Rule**
$$2t_{\mathrm{mix}}^{\pi} \geq D^{\pi} \geq D_x^{\pi} \geq D_z^{\pi}$$

Figure 1: **Continual RL Setting:** The state space $s \in \mathcal{S}$ is decomposed as $s = [x, z]$ where $z \in \mathcal{Z}$ is the task and $x \in \mathcal{X}_z$ is the within-task state. The mixing time, $t_{\mathrm{mix}}^{\pi}$, is lower-bounded by the diameter over the full state space $D^{\pi}$. The latter is lower-bounded by the diameter over the within-task state space $D_x^{\pi}$, which in turn is lower-bounded by the diameter over the space of tasks, $D_z^{\pi}$.

times and continual RL: task locality and bottleneck structure inherently lead to environments with correspondingly high mixing times due to high minimum diameters between states in different tasks.

## 3 Scalable MDPs

In this section, we formalize the notion of scaling in the context of MDPs and provide intuition about the effect of this scaling with the aid of the schematic shown in Figure 2. We assume MDPs can be described in terms of an $n$-dimensional parameter vector $\boldsymbol{q} \in \mathbb{R}^n$, and that some subset of these dimensions are the parameters of interest that will be scaled up. We would like to analyze the effect of this scaling on mixing times.[3] With reference to Figure 1, these parameters can be non-spatial such as the number of tasks, or spatial such as the size of an individual task. More generally and with reference to Figure 2, the effect of scaling up parameters can contribute to longer mixing times by adding bottleneck states (akin to increasing the number of tasks) and increasing the size of the regions generated by a bottleneck (akin to increasing the size of each task). How parameters are scaled is specified by a scaling function, $\sigma$ controlled by scaling parameter, $\nu$.[4] All MDPs accessible through $\sigma$ by varying $\nu$ form a family of MDPs, $\mathbb{C}_\sigma = \{\mathcal{M}_\nu\}$, where $\mathcal{M}_\nu$ is the MDP specified by $\boldsymbol{q}_\nu$. In particular, we consider *proportional scaling* functions, for which a subset of elements of $\boldsymbol{q}$ are scaled linearly with $\nu$. In our experiments, we show results for proportional scaling of two MDP parameters central to continual RL: the number of tasks and the time between task switches.

As the MDP scales up according to $\sigma$, the overall state space size will grow with $|\mathcal{S}| \to \infty$ as $\nu \to \infty$. Regions within the state space can also grow and the steady-state probability on them from $\pi$ may change as a result. A region $\mathcal{R} \subseteq \mathcal{S}$ is a connected subset of states with steady-state probability $\mu^\pi(\mathcal{R}) = \sum_{s \in \mathcal{R}} \mu^\pi(s)$. The boundary of $\mathcal{R}$ is a subset $\partial\mathcal{R} \subseteq \mathcal{R}$ with states having finite probability of transitioning to at least one state that is outside of $\mathcal{R}$, $\sum_{s' \in \mathcal{S} \setminus \mathcal{R}} T^\pi(s'|s) > 0 \quad \forall s \in \partial\mathcal{R}$. We denote the scalable regions and boundaries $\mathcal{R}_\nu$ and $\partial\mathcal{R}_\nu$, respectively. If the size of a region's boundary grows faster than the region's interior, there is a finite state space size at which the region no longer has an interior. In that case, the problem type is of bounded complexity and not relevant to the development of scaling friendly algorithms. We are thus interested in problems where scalable regions can maintain an interior as they are scaled (as measured by steady-state probability) in the limit of large $\nu$. See Appendix D for a precise formulation leading to the following definition:

**Definition 1** *A **scalable MDP** is a family of MDPs $\mathbb{C}_\sigma = \{\mathcal{M}_\nu\}$ arising from a proportional scaling function $\sigma$ satisfying the property that there exists an initial scalable region $\mathcal{R}_0$ with finite interior, $\mu^{\pi^*}(\partial\mathcal{R}_0) < \mu^{\pi^*}(\mathcal{R}_0)$, that scales so that $\mu^{\pi^*}(\partial\mathcal{R}_\nu) < \mu^{\pi^*}(\mathcal{R}_\nu)$ as $\nu \to \infty$ and thus $|\mathcal{S}| \to \infty$.*

---

[3]Scalar parameters with discrete domains are incorporated by embedding them into $\mathbb{R}$.

[4]Formally, this is a scaling deformation, $\sigma : \mathbb{R}^n \times \mathbb{R} \to \mathbb{R}^n$ parameterized by an order parameter $\nu \in \mathbb{R}$ that takes any $\boldsymbol{q}_0$ to $\boldsymbol{q}_\nu = \sigma(\boldsymbol{q}_0, \nu)$, with $\sigma(\cdot, 0)$ as the identity map. Proportional scaling has $q_{\nu,i} \propto \nu$ for all $i$ indexing scaled parameters such that $\boldsymbol{q}_\nu = \boldsymbol{q}_0 + \nu\Delta\boldsymbol{q}$, with $\Delta\boldsymbol{q} \in \mathbb{R}^n$ giving the rates of linear growth in $\boldsymbol{q}$ with $\nu$ up from $\boldsymbol{q}_0$.

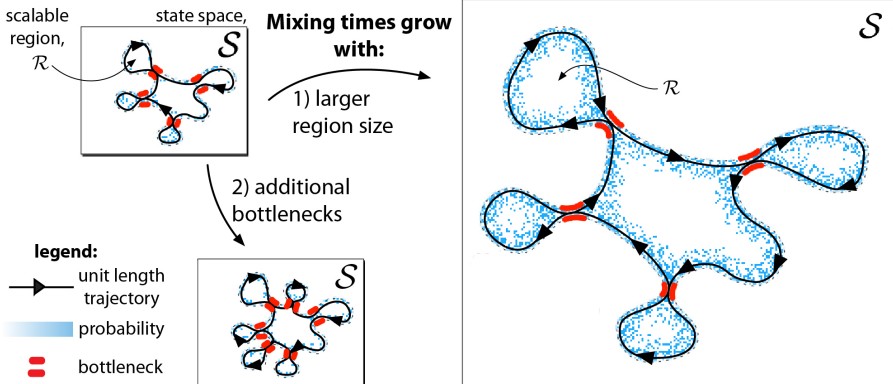

Figure 2: **Mixing times grow as MDPs are scaled up.** *Top left*: A more general version of the continual RL setting shown in Figure 1, where individual tasks correspond to regions, $\mathcal{R}$, of the state space, $\mathcal{S}$, connected through bottlenecks (see legend). An example of the possible steady-state probability of $\pi^*$ is shown in black gradient. The mixing time of an MDP can grow by increasing its diameter (number of equidistant arrow heads) via (1) scaling that increases the size of visited regions $\mathcal{R}$ and thus the expected residence time $t_{\mathcal{R}}^{\pi^*}$ of $\pi^*$ (right) and/or (2) scaling that increases the number of bottlenecks between regions of the state space (bottom left).

## 4  Polynomial Mixing Times

Prior work analyzing the scaling of mixing times makes the practical distinction between *slow mixing* Markov chains that scale exponentially with a size parameter and *rapid mixing* Markov chains that scale at most polynomially. Since the learning speed of any algorithm is lower bounded by the mixing time of the optimal policy [9], environments with a slow mixing Markov chain induced by the optimal policy may very well be outside the practical reach of RL algorithms. In this section, we would like to further characterize mixing times over the space of practically solvable problems of sufficient complexity to still be of interest for real-world applications.

For continuing environments, the tightest known lower bound satisfying Assumption 1 has $H$-step regret $\text{Regret}(H) \in \Omega(\sqrt{D^{\pi^*}|\mathcal{S}||\mathcal{A}|H}) \subseteq \Omega(\sqrt{D^*|\mathcal{S}||\mathcal{A}|H})$ [11].[5] Additionally, because we know that $D^* \geq \log_{|\mathcal{A}|}(|\mathcal{S}|) - 3$ [11], we can bound our regret in the general case as $\text{Regret}(H) \in \tilde{\Omega}(\sqrt{|\mathcal{S}||\mathcal{A}|H})$. Few practical problems are simple enough to exhibit diameters with such logarithmic contributions. More typical and thus of greater interest are diameters (and mixing times) having polynomial scaling in $|\mathcal{S}|$. Focusing on polynomial scaling is thus warranted, and provides more stringent lower bounds on regret. We thus consider the following definition:

**Definition 2** *A set or family of MDPs $\mathbb{C}$ has a **polynomial mixing time** if the environment mixing dynamics contributes a $\Omega(|\mathcal{S}|^k)$ multiplicative increase for some $k > 0$ to the intrinsic lower bound on regret as $|\mathcal{S}| \to \infty \ \forall \mathcal{M} \in \mathbb{C}$.*

An immediate utility of Definition 2 is that it subsumes the diversity of different diameter and mixing time definitions by explicitly stating their equivalence in scaling with respect to the state space size:

**Proposition 1** *If all MDPs $\mathcal{M}$ within the subclass of MDPs $\mathbb{C}$ have $t_{\text{ret}}^{\pi^*}$, $t_{\text{ces}}^{\pi^*}$, $t_{\text{mix}}^{\pi^*}$, $D^{\pi^*}$, or $D^* \in \Omega(|\mathcal{S}|^k)$ for some $k > 0$ we can say that $\mathbb{C}$ has a polynomial mixing time.*[6]

We can thus hereon focus on formulating scaling in MDPs through their state space size. One way to understand how a scalable region, $\mathcal{R}_\nu$, contributes to the mixing time is through its *residence time*, $t_{\mathcal{R}}^{\pi^*}$, *i.e.* the average time that $\pi^*$ spends in $\mathcal{R}$ during a single visit. Using results from the theory of bottleneck ratios in Markov chains [10], we show that if scaling the MDP leads to $t_{\mathcal{R}}^{\pi^*}$ increasing by a polynomial factor in $|\mathcal{S}|$, then the mixing time also increases at a polynomial rate:

---

[5]The bias span is only lower for weakly communicating MDPs violating Assumption 1 so that $D^* = \infty$.

[6]The $\epsilon$-return mixing time $t_{\text{ret}}^{\pi}(\epsilon)$ is for assessing mixing from the perspective of accumulated rewards, whereas the $\epsilon$-Cesaro mixing time $t_{\text{ces}}^{\pi}(\epsilon)$ is for periodic problems that do not converge in the limit from Corollary 1. See Appendix D for additional details.

**Proposition 2** *Any scalable MDP $\mathbb{C}_\sigma$ exhibits a polynomial mixing time if there exists a scalable region $\mathcal{R}_\nu$ such that $\mathbb{E}_{\mu^{\pi^*}}[t_{\mathcal{R}_\nu}^{\pi^*}] \in \Omega(|\mathcal{S}|^k)$ for some $k > 0$.*

The purpose of Proposition 2 and Figure 2 are to provide intuition to readers about why scalable MDPs inherently must have polynomial mixing times. Our paper's main result builds off Proposition 2, to make a general statement about the set of all possible scalable MDPs:

**Theorem 1** *(Mixing Time Scaling): Any scalable MDP $\mathbb{C}_\sigma$ has a polynomial mixing time.*

## 5  Myopic Bias During Scaling

Monte Carlo sampling and bootstrapping are the two primary policy evaluation frameworks for RL [5]. Both implicitly assume that a finite and fixed maximum frequency $f^*(\pi) \in [0,1]$ of policy improvement steps is possible for unbiased updates regardless of the problem size. We now propose three corollaries of Theorem 1 that together argue how polynomial mixing times invalidate this assumption simply because $f^*(\pi) \geq 1/t_{\text{mix}}^\pi$, and $t_{\text{mix}}^\pi$ grows over scalable MDPs. In practice, algorithm designers cannot afford to wait until reaching the mixing time before updating their model when mixing times are high. This results in a myopic bias in the policy improvement steps taken.

**Monte Carlo sampling:** In continuing environments with polynomial mixing times, this sampling procedure poses a problem when optimizing for $\rho(\pi)$. To get an unbiased estimate of $\rho(\pi)$, we must be able to sample from the steady-state distribution $\mu^\pi(s)$, which is only available after $t_{\text{mix}}^\pi$ steps in the environment. Moreover, as demonstrated by Zahavy et al. [13] in the general case where no upper bound on the mixing time is known apriori, $O(|\mathcal{S}|t_{\text{mix}}^\pi)$ samples are needed to retrieve a single unbiased sample from $\mu^\pi(s)$. As such, it is clear that the length of the policy evaluation phase strongly depends on the mixing time of the current policy $\pi$ and thus the maximum frequency $f^*(\pi)$ of unbiased policy improvement steps decreases as the mixing time increases:

**Corollary 2** *A Monte Carlo sampling algorithm for policy $\pi$ in scalable MDP $\mathbb{C}_\sigma$ has a maximum frequency of unbiased policy improvement steps $f^*(\pi) \to 0$ as $|\mathcal{S}| \to \infty$.*

For scalable MDPs of significant size, Corollary 2 implies that model-free Monte Carlo methods perform unbiased updates arbitrarily slowly and that model-based Monte Carlo methods will need arbitrary amounts of compute for unbiased updates even when a true environment model is known. As such, to address scaling concerns over large horizons, bootstrapping methods based on the Bellman equation are generally recommended [5].

**Bootstrapping for Evaluation:** This process is referred to as *iterative policy evaluation* [5] and is known to converge to the true $V^\pi$ in the limit of infinite evaluations of a fixed policy. However, in practice, RL algorithms constantly change their policy as they learn and only can afford partial backups during policy evaluation when applied to large-scale domains. Indeed, a foundational theoretical principle in RL is that as long as the agent constantly explores all states and actions with some probability, bootstrapping with only partial backups will still allow an agent to learn $\pi^*$ in the limit of many samples [5]. However, sample efficiency can still be quite poor since partial backups insert bias into each individual policy evaluation step. This bias, referred to as *staleness* [14], arises when the value function used for bootstrapping at the next state is reflective of an old policy that is currently out of date. Unfortunately, if we want to avoid staleness bias during learning, the length of policy evaluation for each policy must once again depend strongly on the mixing time:

**Corollary 3** *A bootstrapping algorithm with policy $\pi$ in scalable MDP $\mathbb{C}_\sigma$ has a maximum frequency of unbiased policy improvement steps $f^*(\pi) \to 0$ as $|\mathcal{S}| \to \infty$.*

**Bootstrapping for Improvement:** is the process of using bootstrapping for policy improvement as popularized by dynamic programming algorithms such as policy iteration and value iteration, as well as by RL frameworks such as temporal difference (TD) learning and actor-critic. The theoretical foundation of these approaches is the *policy improvement theorem*, which demonstrates the value of taking a greedy action with respect to the estimated action-value function $Q^\pi(s,a)$ at each step. The policy improvement theorem then tells us that policy changes towards $\pi'$ are worthwhile as $V^\pi(s) \leq Q^\pi(s, \pi'(s)) \leq V^{\pi'}(s)$ eventually yielding $\pi^*$ in the limit of many changes [5]. However, this improvement is unlikely to be efficient when mixing times are high. To demonstrate this, we decompose value into the transient and limiting components respectively $V^\pi(s) = V_{\text{trans}}^\pi(s) + V_{\text{lim}}^\pi(s)$. While clearly $V^\pi(s) \leq Q^\pi(s, \pi'(s))$, the functions both follow $\pi$ into the future, so their limiting distribution is the same implying that $V_{\text{trans}}^\pi(s) \leq Q_{\text{trans}}^\pi(s, \pi'(s))$ and $V_{\text{lim}}^\pi(s) = Q_{\text{lim}}^\pi(s, \pi'(s))$.

**Corollary 4** *Policy improvement steps with bootstrapping based on the Bellman optimality operator guarantee monotonic improvement for $V_{\text{trans}}^\pi$, but do not for $V_{\text{lim}}^\pi$.*

See Appendix A for details on how these ideas connect to relevant off-policy or offline RL approaches and to the literature on catastrophic forgetting in continual RL.

# 6 Empirical Analysis of Mixing Behavior

We have established theoretically that scalable MDPs have polynomial mixing times and that polynomial mixing times present significant optimization difficulties for current approaches to RL. However, we still must demonstrate 1) that scalable MDPs are a useful construct for understanding the scaling process within modern continual RL benchmarks and 2) that large mixing times become a significant practical impediment to performing reliable policy evaluation in these domains. Towards this end, we consider empirical scaling of the mixing time with respect to the number of distinct tasks, $|\mathcal{Z}|$, and to the task duration, $\tau$, which controls the bottleneck structure. For this purpose, we evaluate a set of high-quality pretrained policies. In both cases, we report empirical results via the dependence of the $\epsilon$-return mixing time (equation 2) on the relative precision with which the reward rate is estimated. Presenting the relationship in this way ensures we isolate the contribution that myopic policy evaluation has on the performance relative to the optimum and that high mixing times are not inflated by spurious sources of distributional mismatch that do not hinder policy evaluation.

**Empirical Mixing Time Estimation.** In order to quantitatively estimate the $\epsilon$-return mixing time $t_{\text{ret}}^\pi(\epsilon)$ in equation 2, we need to estimate two terms: the average true reward rate $\rho(\pi)$ (which is agnostic to the start state), and the $h$-step undiscounted return $\rho(h, s_0, \pi)$ from a start state $s_0$ for all $s \in \mathcal{S}$. We can then provide $t_{\text{ret}}^\pi(\epsilon)$ for any desired value of $\epsilon$. To calculate $\rho(\pi)$ we unroll the policy $\pi$ in the environment for a large number of time steps (more than a million steps) while accumulating the rewards that we received and finally compute the average over the total number of environment interactions. Estimating $\rho(h, s, \pi)$ for every start state $s \in \mathcal{S}$ in equation 2 is challenging because of the large state spaces in Atari and Mujoco. Hence we rely on approximations where we choose a subset of states from $\mathcal{S}$ to estimate $\rho(h, s, \pi)$. However, since the estimates could be biased based on the start states we choose, instead of randomly sampling a fixed set of start states, we leverage reservoir sampling [15] to ensure that the limited fraction of possible start states that we consider is unbiased according to the on-policy distribution. Conservative definitions of the mixing time take the maximum over start states [9]. Here, since we are interested in the characteristic mixing time size, we report the average among start states, which ensures that the mixing times reported are representative of those actually needed for evaluation over the course of on-policy training. Finally, in order to provide an intuitive interpretation for $\epsilon$ across domains, we leverage the *relative error*, $\epsilon/\rho(\pi)$, as a common reference point for comparison. We can then calculate the $\epsilon$-return mixing times for a fixed relative error. See Appendix B for further details.[7]

**Setting of Interest.** We focus our experiments on the following scalable MDP formulation that is broadly representative of the majority of work on continual RL [16, 17, 18, 19, 20]:

**Example 1** *(Continual Learning with Passive Task Switching): Consider an environment with tasks $z \in \mathcal{Z}$ and within-task states $x \in \mathcal{X}_z$. The residence time of the region encompassed by task $z$ before moving to another region is fixed for all tasks at $\tau$ for any $\pi$ such that $t_{\mathcal{X}_z}^\pi = \tau \;\; \forall z \in \mathcal{Z}$. Regardless of the way that tasks are connected, the diameter of such an MDP must scale as $D^* \in \Omega(\tau|\mathcal{Z}|)$ because the residence time bounds the minimum possible time to travel between two states in different tasks. Utilizing our formulation of scalable MDPs with initial MDP parameters $\boldsymbol{q}_0 = (\tau, |\mathcal{Z}|)$, the space of possible proportional scaling functions on $\tau$ only, $|\mathcal{Z}|$ only, and both $\tau$ and $|\mathcal{Z}|$ simultaneously are $\sigma_\tau(\boldsymbol{q}_0, \nu) = (\tau + a\nu, |\mathcal{Z}|)$, $\sigma_{|\mathcal{Z}|}(\boldsymbol{q}_0, \nu) = (\tau, |\mathcal{Z}| + b\nu)$, and $\sigma_{\tau, |\mathcal{Z}|}(\boldsymbol{q}_0, \nu) = (\tau + a\nu, |\mathcal{Z}| + b\nu)$, respectively, where $a, b \in \mathbb{R}$ span the respective spaces.*

## 6.1 A Simple Motivating Example

We consider a simple $10 \times 10 \times 10$ 3-dimensional grid world environment with 6 actions corresponding to up and down actions in each dimension. In this environment, we can define 1,000 different tasks corresponding to considering each unique state as a goal location. At each time-step the agent is sent

---

[7]Our code is available at https://github.com/SharathRaparthy/polynomial_mixing_times.git.

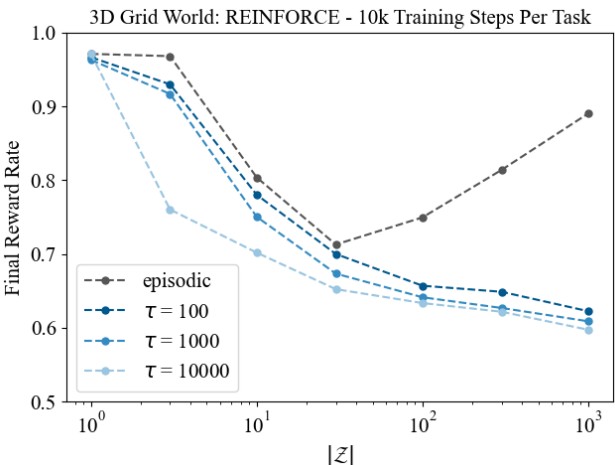

Figure 3: Average reward rate across tasks at the end of training as a function of $|\mathcal{Z}|$ and $\tau$.

a 60-dimensional concatenation of six 10-dimensional one-hot vectors. The first three one-hot vectors correspond to the agent's x,y, and z coordinates and the second three correspond to the goal location.

**Myopic Optimization Bias:** As we have motivated in Section 2.2, this environment is made to demonstrate the difficulty associated with myopic optimization bias even when task boundaries and their relation are fully observable. We consider a continual episodic RL formulation following the description in Example 1 where task sequences arise from changing the goal location every $\tau$ steps and the set of goal locations defines the set of tasks, $\mathcal{Z}$. As such, the diameter of the MDP clearly increases as $\tau$ and the number of tasks $|\mathcal{Z}|$ are scaled up, leading to at least linear scaling. As a result, an agent that updates only based on the current episode experiences myopic bias with respect to the steady-state distribution over which the tasks are uniformly balanced.

**Training Procedure:** The 60-dimensional sparse observation vector is processed by a multi-layer perceptron with two 100 unit hidden layers and Relu activations. The agent performs optimization following episodic REINFORCE. We explore different configurations of this scalable MDP by varying $|\mathcal{Z}|$ from 1 to 1,000 and varying $\tau$ from 100 to 10,000. An important baseline to consider is the extreme of switching tasks after every episode as in multi-task RL (which is generally considered an upper bound on continual RL performance). The agent receives a reward of 1 every step it makes progress towards its goal and a reward of 0 otherwise. This implies that the reward rate of $\pi^*$ is $\rho(\pi^*) = 1$ and that the reward rate of a uniform action policy is 0.5 for every task $z \in \mathcal{Z}$ regardless of the distance travelled from the start state for each episode. For each $|\mathcal{Z}|$ and $\tau$, we train the agent for 10,000 total steps and report the average reward rate after training as an average across 300 seeds, which correspond to randomly selected task progressions. See Appendix C for further details.

**Results:** We display the results of these experiments in Figure 3. It is interesting to notice the episodic transition performance, which can near flawlessly learn a single task in 10,000 steps but struggles significantly with interference in the multitask setting. Performance goes down from 1 to 30 tasks and appears to rebound afterward, which is logical because the similarity between incoming tasks and old tasks is increasing and the total number of training steps across tasks is going up. When both $\tau$ and $|\mathcal{Z}|$ are high at the same time, learning performance is quite poor (not much better than a uniform policy). The starting location is always in the same corner, so some commonalities can be exploited even by a policy experiencing significant interference. Performance bottoms out after fewer tasks when $\tau$ is larger, which makes sense as that is when the myopic bias of optimization is greatest.

## 6.2 Overview of Empirical Findings on Atari and Mujoco

**Domains of Focus.** We perform experiments involving sequential interaction across 7 Atari environments: *Breakout, Pong, SpaceInvaders, BeamRider, Enduro, SeaQuest* and *Qbert*. This is a typical number of tasks explored in the continual RL literature [16, 17, 18, 19, 20]. We consider a sequential Atari setup where the individual environments can be seen as sub-regions of a larger environment that are stitched together (see Figure 1 and Appendix A for further details). To demonstrate the validity of our results on vector-valued state spaces, we also consider sequential Mujoco experiments with

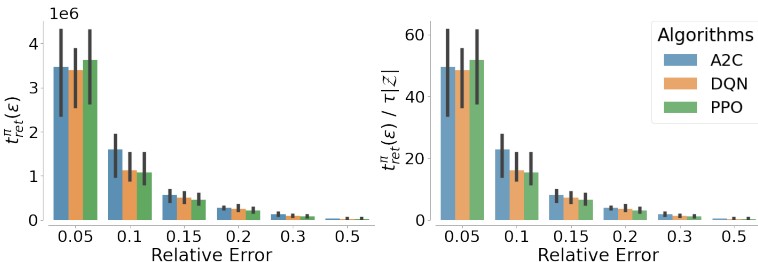

Figure 4: **Mixing time as a function of relative error for fixed scaling parameters.** *Left*: $\epsilon$-return mixing time (equation 2) as a function of relative error, $\epsilon/\rho(\pi)$, in reward rate estimation for 3 standard algorithms. *Right*: Same as left, here normalized by $\tau|\mathcal{Z}|$ where $\tau = 10,000$ and $|\mathcal{Z}| = 7$. Note the difference in range on the y-axis, which reflects this normalization.

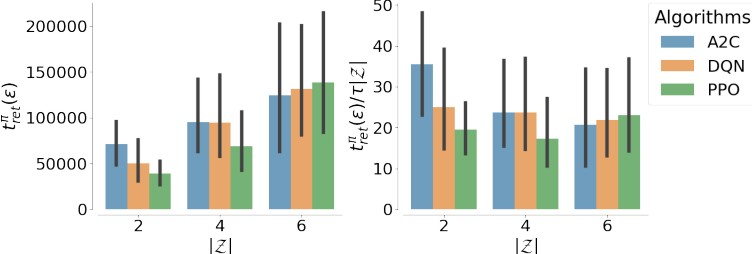

Figure 5: **Mixing time scaling with the number of tasks $|\mathcal{Z}|$.** *Left*: Average $\epsilon$-return mixing time across different task combinations $\mathcal{Z}$ for different algorithms. *Right*: Same as left, here normalized by $\tau|\mathcal{Z}|$ where $\tau =1,000$. Note the difference in range on the y-axis, which reflects this normalization.

the following 5 environments: *HalfCheetah, Hopper, Walker2d, Ant* and *Swimmer*. For both our sequential Atari and Mujoco experiments, we leverage high-performing pretrained policies that are publicly available [21] for mixing time calculations. We leverage task labels to use the pretrained model specific to each task as our behavior policy as appropriate when tasks change.

**Mixing Time as a Function of Relative Error.** As the task connection structure following Example 1 can take any arbitrary form, we have focused our experiments on random transitions between tasks. In Figure 4 we plot both $t_{\text{ret}}^\pi(\epsilon)$ and $t_{\text{ret}}^\pi(\epsilon)/\tau|\mathcal{Z}|$ for $\tau = 10,000$ and $|\mathcal{Z}| = 7$ as a function of the relative error of $\rho(\pi, s, h)$ with respect to $\rho(\pi)$, i.e. $\epsilon = \rho(\pi, s, h) - \rho(\pi)$. We see the intuitive fact that higher demands for precision define longer mixing times and that it is clear that the lower bound on the diameter is a very conservative estimate of the mixing time where we only see $t_{\text{ret}}^\pi(\epsilon)/\tau|\mathcal{Z}| < 1$ for reward rates estimated to a poor tolerance of more than 30% error.

**Scaling $|\mathcal{Z}|$.** We now analyze Example 1 in the case where the state space and diameter grow as we increase the number of tasks $|\mathcal{Z}|$. Specifically we scale $|\mathcal{Z}|$ from 2 to 4 to 6 as $\tau$ is kept fixed at 1,000 and generate 10 random task combinations from the list of 7 for each value of $|\mathcal{Z}|$. In Figure 5 we plot the resulting mixing times for a precision of 10% error in approximating the reward rate. We find that there is a linear trend in the expected mixing time and not just the theoretical lower bound.

**Scaling $\tau$.** We finally analyze Example 1 in the case where the state space is held constant but the diameter grows with the increasingly severe bottleneck structure as we increase $\tau$. Here we consider $|\mathcal{Z}| = 7$ and $\tau = 1,000, 3,000, 10,000,$ and $30,000$, while calculating $\rho(h, s, \pi)$ and $\rho(\pi)$ by unrolling

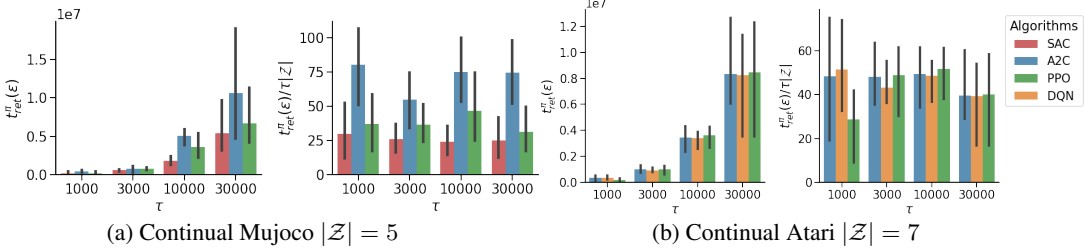

(a) Continual Mujoco $|\mathcal{Z}| = 5$        (b) Continual Atari $|\mathcal{Z}| = 7$

Figure 6: **Mixing time scaling with the task duration, $\tau$.** Left (a, b): $\epsilon$-return mixing time across different $\tau$ values for different algorithms. Right (a, b): Same as left, here normalized by $\tau|\mathcal{Z}|$. Note the difference in range on the y-axis and the distinct subset of algorithms tested (see legend).

the policy $\pi$ for $\tau \times 1{,}000$ environment steps each. In Figure 6 we once again plot both $t_{\text{ret}}^{\pi}(\epsilon)$ and the value normalized by the diameter lower bound $t_{\text{ret}}^{\pi}(\epsilon)/\tau|\mathcal{Z}|$ to find that empirical average mixing times scale linearly with increasing $\tau$ and not just the lower bound.

## 6.3   Implications For Continual RL

In summary, we have observed in Figure 4 that average mixing times reach a staggering high for lower relative errors, and shorter mixing times are associated with very high relative errors. Additionally, in Figures 5 and 6 we observed that, for a broad class of algorithms, there seems to be a linear dependency of the expected mixing time with increasing $|\mathcal{Z}|$ and increasing $\tau$.

**Optimization Instability for Continual Atari.** While we have empirically established that mixing times scale to be quite large in the Atari domain following Example 1 and established theoretically how RL approaches struggle when this is the case in Section 5, we would like to emphasize that the prior literature applying deep RL to continual learning on Atari has empirically come to the same conclusion. Indeed, [17] looked at continual learning across 10 Atari games with a randomized duration between task switches and found that all approaches were significantly outperformed by the single task baseline, reflecting significant instability in optimizing across tasks. Additionally, perhaps the most common setting in the literature considered by [18], [19], and [22] is to train on a set of tasks where $|\mathcal{Z}| = 6$ and $\tau = 50{,}000{,}000$. Note that extrapolating from our results would indicate average mixing times on the order of $\approx 15$ billion steps for a precision of 5% error or $\approx 5$ billion steps for a precision of 10% error. Across a number of RL approaches tried on these domains, as we would expect, all approaches have experienced significant instability in their learning curves. A very popular failure mode is approaches that display too much plasticity such that their performance for a task goes through big peaks and valleys resulting from myopic bias towards optimization on the current task. This phenomena is often known in the literature as catastrophic forgetting [23].

## 7   Inspiring New Approaches for Continual RL

While the focus of our paper is on highlighting the difficulty of the polynomial mixing time problem, not proposing an approach to solve it, we believe that our analysis provides significant insights that the community will be able to draw on to build better approaches for continual RL:

**Direct Steady-State Estimation:** In the context of commonly considered problems that follow the high-level structure of Example 1, we have showcased the deep connection between catastrophic forgetting and myopic bias in the presence of very large mixing times. In Appendix A we demonstrate how this perspective explains some of the value of replay based approaches in this setting. Moreover, this perspective naturally points towards approaches that directly reason over the steady-state distribution as a natural solution to the catastrophic forgetting problem. In Appendix C we consider simple tabular RL experiments across 3 example classes of scalable MDPs. Our analysis demonstrates that on-policy and off-policy versions of Q-Learning that perform policy evaluation using an estimate of the steady-state distribution, derived from matrix inversion of a tabular environment model, consistently outperform standard model-based and model-free baselines in terms of lifelong regret as environments are scaled up. Even though estimating the steady-state distribution would clearly be quite challenging in complex state and observation spaces, recent approaches have made great strides towards developing practical and scalable approaches leveraging data buffers [24].

**Tracking Approaches:** It was originally noted by [25] that it could be a better strategy even in stationary environments with temporal coherence to learn to track the best local solution instead of attempting to converge to a globally optimal fixed policy. This indeed, may be a necessity in environments that experience extreme temporal coherence governed by high mixing times and could provide a formalization of the benefit for approaches based on meta-learning in this context as noted by [25]. For example, [26] recently was successful in addressing the non-stationarity of multi-agent RL by learning to converge to a recurrent set of joint policies rather than a single joint policy.

**Looking Forward.** In this work, we have considered the implicit premise of the current continual RL literature that results from currently manageable small-scale experiments are indicative of performance for the large-scale aspirational use cases of the future. In particular, we have highlighted that mixing times will scale significantly as the problems we deal with are scaled and how traditional approaches to RL are ill-suited to deal with this. We hope that our work will encourage the community to carefully consider the theoretical implications of scaling environments, especially with regard to how proposed continual RL algorithms scale with the mixing time.

## Acknowledgements

We would like to thank Riashat Islam, Gerald Tesauro, Miao Liu and Sarthak Mittal for helpful discussions. Irina Rish acknowledges the support from the Canada CIFAR AI Chair Program and from the Canada Excellence Research Chairs Program. We also thank the IBM Cognitive Compute Cluster, the Mila cluster, and Compute Canada for providing computational resources.

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
