# OpenReview forum: "Continual Learning In Environments With Polynomial Mixing Times"
_NeurIPS.cc/2022/Conference — NeurIPS 2022 Accept_

### Official Review · Reviewer_bA29 · 2022-06-16

**Rating:** 7
**Confidence:** 3
**Soundness:** 3 good
**Presentation:** 3 good
**Contribution:** 3 good

**Summary:**

This paper devises a broad class of MDPs, referred to as Scalable MDPs, for which they provide mathematical evidence to suggest that continual learning algorithms will struggle on. The authors provide several characterizations of the class of Scalable MDPs based on simpler statistical properties of the Markov chains induced by Markov policies. Furthermore, empirical evidence from common RL benchmarks is given to support the hypothesis that continual learning algorithms struggle in Scalable MDPs.

**Questions:**

*Assumption 1*: What is a *unichain* policy? Does this assumption just
mean that all stationary policies induce ergodic Markov chains?

*Corollary 1*: I believe there is a notational mistake here, it says
$\mu^\pi(s) = \lim_{t\to\infty}P^\pi(s_t\mid s_0)$, where the RHS is
completely independent of the LHS. I believe $s_t$ corresponds to $s$
and you're arguing that $s_0$ is irrelevant, but this is a little
confusing.

In the equation under line 81, a similar notational mistake is made,
where I'm assuming $s_h$ corresponds to $s$.

*Section 3*: The definition of a scalable MDP, as well as other concepts
introduced in this section, should be formalized. I also don't find
Figure 2 and its exceptionally long label particularly helpful for
illustrating the concepts.

- *Intrinsic dimensions*: what is the difference between the intrinsic
  dimensions of the state space and the dimensions of the state space
  themselves? Also, who's to say which dimensions should be considered
  extensive rather than intensive? For example, the number of spatial
  dimensions of a gridworld can conceivably be interpreted as both
  extensive and intensive for some tasks, or at least the extensive and
  intensive dimensions can be highly correlated. In this case, I'm not
  sure I understand why it's useful to distinguish them like this.
- *Extensive and Intensive dimensions*: this is more of a nitpick, but
  where do these names come from? They are not at all intuitive to
  me. Maybe external/internal?
- *Scaling functions*: lots of notation is introduced here, and in my
  opinion, it's "over-engineered". It says "... each $\sigma\in\Sigma$
  induces $\mathbb{C}_\sigma$, a $\nu$-parameterized abstract
  class..." -- if the class is parameterized by $\nu$, why is it called
  $\mathbb{C}_\sigma$? Then, it says "An important example of $\sigma$
  is *proportional scaling*, $\sigma(q, \nu) = c(\nu)\circ q$ with
  $c_i(\nu)\in\Theta(\nu)$. There's a lot that confuses me here, namely:
  1. What is $\Theta(\nu)$?
  2. What is the $\circ$ in $c(\nu)\circ q$? Usually $\circ$ refers to
     function composition, but I believe this should actually be
     elementwise multiplication, which I'd expect to be written as
     $c(\nu)\otimes q$.
  3. It's confusing to have $\sigma$ parameterized by a scalar $\nu$
     only to then introduce this new function $c$ with the purpose of
     (to my understanding) transforming $\nu$ into a vector by some
     mysterious process. Why not allow $\sigma$ to be parameterized by
     an arbitrary object and do away with $c$ altogether? This is what I
     meant by "over-engineering", my impression is that there are too
     many "free variables" here than are necessary.
  4. Since ultimately you're going to restrict $\sigma$ to be a
     proportional scaling function, why bother with all of the
     structural conditions you impose on $\sigma$, like $\sigma(\cdot,
     0)$ being the identity? Especially since $\sigma$ will be
     multiplicative, having $0$ represent the identity is awkward and
     all of these conditions just make the presentation much more
     difficult to comprehend.

After having finished section 3, my impression is that all of this
machinery that was introduced will be used simply to encode a scaling
operation on the features of the state space. Is it really necessary to
introduce all of this abstraction?

On line 142, it says

      Applying this distinction to our task formulation and
      example in Figure 1, intensive scaling increases the number
      of tasks, and extensive scaling increases the size of each
      task.

This sentence is much clearer to me than the previous ones discussing
extensive and intensive scaling. What does the remainder of section 3
tell us that this statement doesn't?

*Definition of Scalable MDP*:
- Is $\mathbb{M}$ ever defined? I assume it's the set of all classes of
  MDPs than are induced by scaling functions?
- Why specify that $\sigma$ is a scaling operation on a subset of
  intrinsic dimensions? I think it would be clearer to omit this, since
  one can avoid scaling dimensions $i$ by having $c_i(\nu) = 1$.
- What is an /applicable scalable/ region?
- Why must $\mu^{\pi^*}(\mathcal{R})\leq 1/2$? Is this what defines an
  applicable scalable region? What is the significance of such a region?
- What does it mean for the inequality to be preserved as
  $|\mathcal{S}|\to\infty$? Why/how is the cardinality of the state
  space increasing?

I think it would go a long way to give some intuition for this
definition.

*Definition 2*: Is this definition novel? If so, I think the name should
be changed -- the "polynomial mixing time" is defined here in terms of
regret, and very indirectly w.r.t. the mixing time.

Rather, why not define polynomial mixing time by Proposition 1? For
example, "a family of MDPs $\mathbb{C}$ is said to have a polynomial
mixing time if for each $\mathcal{M}\in\mathbb{C}$ we have
$\min(t_{\text{ret}}^{\pi^*}, t_{\text{ces}}^{\pi^*},
t_{\text{mix}}^{\pi^*}, D^{\pi^*}, D^*)\in\Omega(|\mathcal{S}|^k)$...".

*Proposition 2*: What are "scalable regions"? I'm assuming this is the
same as the "applicable scalable regions" from before, but then it
doesn't help to specify that the MDPs have scalable regions /and/ that
there exists a region $\mathcal{R}$ with $\mu^{\pi^*}(\mathcal{R})\leq
1/2$. As I understand it, Proposition 2 claims that a family of MDPs has
a polynomial mixing time if it has a scalable region with polynomial
residence time.

Right before Theorem 1, it says

      The purpose of Proposition 2 and Figure 2 are to provide
      intuition to readers about why scalable MDPs inherently must
      have polynomial mixing times.

At least personally, I find Figure 2 to be much too dense and
mysterious. For example, why are $\mathcal{R}$ and
$\mu^{\pi^*}(\mathcal{R})$ pointing to two distinct areas always?  Also,
what are the translucent blue "shadows" around the black paths?  How
does the number of equidistant arrowheads reflect the residence time?

*Section 5*: Why is this called "myopic" bias? The word "myopic" occurs
only in the section title. Also, about the Corollaries, I believe the
same results would hold for arbitrary families of MDPs (since
$t_{\text{mix}}\in \Omega(\log|S|)$, so $f^*(\pi)\in o(1)$).

Moreover, most of the content in this section is background material for
value-based RL methods, and has nothing to do with continual learning
and/or mixing times. I think it would be better to move such content to
the background section or to the appendix, as it disturbs the flow of
logic in the paper.


**Limitations:**

As mentioned, I believe the method of empirically validating the theoretical asymptotic results has limited value.

Moreover, the proofs of just about all of the theoretical claims were omitted, which limits the soundness of the paper. Having said that, I believe some of the definitions presented in this work are valuable contributions on their own.

**Strengths And Weaknesses:**

# Strengths

I believe the direction of work chosen in this paper -- that being the goal of classifying families of MDPs by complexity in certain regimes -- to be very interesting and important. The presentation of the paper is nice and the grammar is mostly quite good.

# Weaknesses

Unfortunately, I find much of the mathematical formalism to be highly confusing and convoluted. I believe many of the abstractions presented in the paper are unnecessarily complicated, making it difficult to parse some sections of the paper. Also, I found that very little intuition was given for the definitions that were contributed in this paper. Figure 2 is extremely difficult to comprehend (at least in my opinion). Finally, I don't quite see the value in the empirical results. It doesn't really make sense to me to estimate asymptotic rates this way, since the number of tasks ranges only from 1 to 7. The empirical results do not necessarily suggest any asymptotic trend. While I agree that those inferred by the authors are fairly reasonable, they're also not very surprising, so I don't find the empirical evidence helped convince me of any properties of Scalable MDPs.

Also, it appears that the authors submitted a copy of the paper in place of the appendix, so I could not find any of the proofs. This is really a shame, since I was hoping that reading through the proofs would help me understand the intuition and rationale behind some of the contributions of this work.

Due to the lack of proofs as well as what I understand to be a weak method of empirical validation, I have to give this a score of 3.

### Post-Rebuttal Edit

The revised version of the paper is, in my opinion, quite a bit more clear and easy to read. I still believe Figure 2 can be improved, since I find it difficult to infer from the diagram what is said in its caption. In general, the authors addressed most of my concerns. In particular, the simplified mathematical framework is tremendously more legible (to me) than the original one.

Upon reading the proofs, the most I can say is that they look reasonable and I couldn't find any clear flaws. Having said that, some of the proofs allude to theorems from the literature that I'm not very familiar with, and I have not had as much time as I would have liked to dive deeper into these.

Overall, I do think this paper presents an interesting approach to understanding the complexity of continual RL. I have increased my score to reflect the clarifications in the revised paper as well as the inclusion of the proofs.

---

> ### Author Response · Authors · 2022-08-02
> **Author Response**
>
> We want to begin by expressing our immense gratitude to you for providing such a thoughtful and detailed review of our work. We know how overworked reviewers are at these conferences and we really appreciate the exceptionally detailed nature of your feedback. We still hope to convince you that our paper would make a nice contribution to this conference, but feel that either way our paper’s presentation has benefited greatly from our attempts to address your detailed reading of it.
>
> **Revised Appendix:**  Thank you so much for bringing the unfortunate situation about the appendix in our submission to our attention. We were mortified to learn about this mistake during our submission. We have now included the actual appendix in the revision. We know there is little we can do to take back the initial error that we carelessly made. We hope that it is still possible for our rejection or acceptance to be decided on a substantive basis and will focus primarily on addressing your other high-level concerns. We would be more than happy to have a back and forth during the discussion period about any questions you have regarding the proofs in the appendix.
>
> **Addressing High-Level Feedback:** Thank you for your kind words regarding the value of our theoretical contributions and the impact of the direction we have chosen to explore. We understand that you primarily have the following high-level concerns about our submission:
> * **The mathematical formalism is confusing and convoluted:** We highly appreciate your detailed comments in this regard and have addressed each case specifically below. Please let us know if we can provide additional clarity with respect to any of these points.
> * **Weak Asymptotic Results:** Other reviewers mentioned the idea of beginning the experiments with a simple example domain, which we have provided in our comment titled "Motivating Experiments". In this synthetic domain we have been able to simulate up to 1,000 tasks. We hope this helps alleviate your concern about extrapolating with respect to tasks (which are inherently limited in the Atari domain). We have also provided additional experiments in Figure 5 of Section 6.1 demonstrating that scaling properties remain similar to Atari when exploring domains from Mujoco. Please let us know whether this addresses your concern. We are not sure if we fully understood your comment about how the results aren't surprising. Is this because the empirical results naturally follow from the theory presented? Do you have any specific ideas in mind about an experiment we could provide that would be more meaningful to you?

---

> > ### Author Response · Authors · 2022-08-02
> > **Author Response [Part 2]**
> >
> > We will now go on to address the specific questions and feedback you provided directly (areas of the document containing changes are highlighted in blue):
> > * **Unichain Policy:** Yes. You are correct. We were following notation from (Berstekas, 1998), (Wan et al., 2021) and trying to differentiate from the multi-chain case as discussed in (Mahadevan, 1996). This is called an ergodicity assumption in the textbook by Sutton and Barto. We have added a footnote to clarify in the revision based on your comment.
> > * **Notation for Corollary 1 and $\epsilon$-Mixing Time:** Thank you for mentioning this confusion and please see our update in the revision.
> > * **Lack of Clarity in Figure 2:** We have substantially simplified figure 2 to emphasize only the two types of scaling that we consider in this work: 1) scaling the number of tasks and 2) scaling the amount of time spent in each task. The caption is now substantially shorter and more concise. Please let us know if there are additional points about the figure that requires more clarity.
> > * **Lack of Clarity in Section 3:** We have overhauled the section introducing scalable MDPs by summarizing in plain English and moving all but essential formal properties to the appendix. In it's place is what we hope the reviewer finds a more intuitive and direct discussion of the important aspects of our definition. The changes include:
> >   * **Removed ambiguous terminology:** Removing the term **"intrinsic"** to describe the parameters of the MDP which seemed to lead to confusion.
> >   * **Replacement for 'extensive/intensive' terms:** Use of intuitive terms like 'spatial/non-spatial' instead of 'extensive/intensive'. The latter terminology was borrowed from system scaling theory in statistical physics ('extensive' thermodynamic parameters scale with system size, while 'intensive' parameters do not), but admittedly it does not add clarity here.
> >   * **More direct scaling function formulation:** We agree that the function $c$ is not necessary in a minimal description of the idea for readers. Regarding $\circ$-notation, element-wise multiplication unfortunately has multiple symbol conventions. We had used Hadamard product notation $\circ$, which you noted is ambiguous with function composition notation. Looking more into the matter, it seems $\odot$ offers an unambiguous solution.
> >   * **Emphasis on Proportional Scaling:** We have also tried to put more emphasis directly focussing on proportional scaling with fewer potentially distracting details to address your confusion.
> >   * **Removed redundant MDP class notations:** Based on your comment, we have concluded that the notation $\mathbb{M}$ was causing more confusion than clarity and hence we removed any mention of it in the revision.
> >   * **Clarifying why $\mu^{\pi}(\mathcal{R}) < 1/2$:** Thank you for mentioning your confusion regarding the statement of the steady-state probability staying less than or equal to 1/2 in our submitted draft. We have realized that it is not necessary to include this in Definition 1 and Proposition 2 in the main text. As such, we have removed it for clarity in the revision. The 1/2 originates from our usage of Theorem 7.4 from (Levin and Peres, 2017), which relies on the conventional mixing i.e. $t^\pi_{\textrm{mix}}(1/4)$ as a convention. In our updated proof of Proposition 2 we have realized we can easily extend their proof to an arbitrary $\epsilon$ and in doing so have no reason to bog down readers with discussion of upper bounds on the region size.
> >   * **Clarity on Scalable Regions:** We have been updated the document to be more concrete about the definition of a scalable region $\mathcal{R}_\nu$.
> >   * **Clarity on Definition 1:** We have also updated Definition 1 to be more concrete that the size of the state space increases because $\nu$ is increased.

---

> > > ### Author Response · Authors · 2022-08-02
> > > **Author Reponse [Part 3]**
> > >
> > > * **Focus on Regret in Definition 2:** To define a polynomial mixing time, we must by necessity provide some mixing time related metric that can be used to ground the definition. It seems you are suggesting it should be grounded in some set of well known types of mixing times? To us this would seem like an arbitrary choice, whereas impact on regret is directly related to the difficulty of the problem class itself. Notice that this choice essentially boils down to choosing $D^{\pi^*}$ to be the reference mixing time and then proving that if any of the other popular mixing time metrics we consider grow polynomially in the state space it also implies that $D^{\pi^*}$ grows polynomially with the state space. We maintain that it makes sense to call this a polynomial mixing time since if this condition is true, it indeed implies that many popular mixing time related metrics we are aware of grow polynomially in the state space size for the optimal policy. An additional utility of our definition approach is that it provides a means of expanding the set of metrics in Proposition 1 beyond those that we have considered. For example, future work may be interested in the cover time or some other metric we have not considered. There is now a straightforward way that they can prove whether this metric implies polynomial mixing.
> > > * **Comments on Section 5:** Thank you for highlighting that we could provide more clarity about the connection between the update frequency scaling and myopic bias. We have added a couple sentences to make the connection more explicit at the end of the first paragraph of the section. We agree that in our submission we included some background material about general RL that is likely not necessary for our paper's audience. We have attempted to take out some of this excess content in the introduction to each corollary. Please let us know if there is any additional content you were thinking of that isn't necessary.
> > >
> > > Finally we again want to thank you for a thoughtful and thorough review. We tried our best to address each of your concerns. If there are any additional concerns we are more than happy to discuss those during the discussion phase.

---

> > > > ### Author Response · Authors · 2022-08-08
> > > > **Have the revisions addressed your concerns?**
> > > >
> > > > Thank you again for your very thorough review. In our rebuttal, we have addressed your concerns regarding
> > > >
> > > > 1) clarity of the mathematical formalism of scalable MDPs
> > > > 2) asymptotic results for mixing times when scaling the number of tasks by providing results on a simple task (please refer to this plot [https://i.imgur.com/YtYZFlC.png](https://i.imgur.com/YtYZFlC.png)).
> > > >
> > > > Since the discussion phase is closing soon, we would like to know if you have any remaining concerns that we haven't addressed. We will be more than happy to clarify.
> > > >
> > > > Thanks

---

> > > > > ### Comment · Reviewer_bA29 · 2022-08-08
> > > > > **Sorry for the delay**
> > > > >
> > > > > Thanks to the authors for such a thorough response. Evidently none of the other reviewers attempted to read the proofs originally, so I have been going over the appendix carefully. Please excuse the delay. In the meantime, I would like to thank the authors as well for putting in the effort to clear up some of the points of confusion in the revised version of the paper, which I personally believe to be a major improvement.
> > > > >
> > > > > For the record, I believe the original omission of the appendix was an honest mistake, as I find it unlikely that the authors managed to write the recently added appendix in such a short period of time. I will be updating my score shortly, once I've finished going over the appendix.

---

### Official Review · Reviewer_VEPZ · 2022-07-03

**Rating:** 5
**Confidence:** 2
**Soundness:** 3 good
**Presentation:** 2 fair
**Contribution:** 3 good

**Summary:**

In this paper, the authors discuss bounds on the mixing time for the continual RL setting. In particular, the authors raise the setup of scalable MDPs. The authors further characterize the condition for families of MDPs to have polynomial mixing time and show that the scalable MDPs have polynomial mixing time, which guarantees a tight lower bound of the corresponding regret. The authors then discuss the myopic bias that arises for the scalable MDPs with polynomial mixing time. Finally, the authors conduct experiments on Atari to justify their theoretical findings.

**Questions:**

-The setup of continual RL seems to be confusing to me. Given that the transition of states still follows a (static) probability function both across tasks and within tasks, what makes it different between a regular MDP and the continuing environment?

-The setup of scalable MDPs seems to be a bit confusing to me as well. How does the raised expansive property restrict the family of scaling functions and why is it necessary? Could the authors provide more intuition in the definition of scalable MDPs (Definition 1)? Could the authors raise a more concrete example for better understanding? For instance, simple tabular MDPs with explicitly specified transition dynamics, intrinsic variable $q$, and the scaling functions $\Sigma$ are sufficient to demonstrate. In addition, the results could be greatly strengthened if the authors could justify the generality of such scalable MDPs in capturing real-world applications.

-In terms of the discussion of myopic bias, the authors discuss the effect of mixing time on the sample efficiency of policy evaluation. Does similar arguments also hold for value-based approaches such as the fitted-Q-iteration (FQI)? In the typical previous analysis, FQI hinges on good coverage of (notions of) steady-state probability of optimal policy (e.g.[1, 2]). Does such coverage presume tractable mixing time for the policies that collect the data?

-For the experiment, it would be better if the authors could explain why the selected tasks with the scaling parameters fit into the definition of scalable MDPs in Section 3.



[1] Chen and Jiang. Information-Theoretic Considerations in Batch Reinforcement Learning. 2019.

[2] Xie et al., Bellman-consistent pessimism for offline reinforcement learning. 2021.

**Limitations:**

N.A.

**Strengths And Weaknesses:**

Strength:

-The mixing time is scarcely explored yet crucial in the analysis of MDP. The analysis and the perspective of the authors are novel to me. In addition, the theoretical findings are backed up by experiments on Atari.

Weaknesses:

-The presentation of the authors makes it unclear whether such analysis captures real-world applications of RL and hinders the usefulness of the results. See Questions for the details.

---

> ### Author Response · Authors · 2022-08-02
> **Author Response**
>
> Thank you for your thoughtful review and for highlighting the novelty of our theoretical contributions and analysis.
>
> **Continual RL vs. Continuing Environments:** Thank you for raising this important question, the connection between these two terms can be a bit complicated as discussed in detail in the survey on continual RL by (Khetarpal et al., 2020). Generally, non-stationary environments can be seen as stationary environments that depend on some unobserved factor i.e. an unobserved task label. That said, the literature on continual learning demonstrates that optimization issues are generally experienced whether task labels are observed or not.  Our analysis of mixing times in this paper can be seen as an attempt to understand this optimization instability as myopic bias even when the overall environment is fully observable and stationary. In this setting, tasks can be seen as regions of the state space stitched together (whether or not it is acknowledged by the environment or algorithm designer). We have positioned our analysis in this way to be as general as possible as considerations in the fully observable case will still be important when partial observability and apparent true system non-stationarity are also inserted into the problem. We have added a couple of sentences to Section 2.2 to provide additional clarity in this regard. In Appendix A.1 we have provided a more comprehensive comparison to the literature on continual RL. Please let us know if there are any elements you still find confusing about our positioning and we would be happy to address them.
>
>
> **Upper Bound on Steady-State Probabilities:** Thank you for mentioning your confusion regarding the statement of the steady-state probability staying less than or equal to 1/2 in our submitted draft. We have realized that it is not necessary to include this in Definition 1 and Proposition 2 in the main text. As such, we have removed it for clarity in the revision. This originated from our usage of Theorem 7.4 from (Levin and Peres, 2017), which relies on the conventional $\epsilon$-return mixing time $t^\pi_{\textrm{mix}}(1/4)$ as a convention. In our updated proof of Proposition 2 we have realized we can easily extend their proof to an arbitrary $\epsilon$ and that after doing so we have no reason to bog down readers with discussion of upper bounds on the region probability density. Please let us know if you continue to find some aspects of Definition 1 confusing or lacking in motivation.
>
> **Intuition of Scalable MDPs:** Thanks for raising this concern and we agree that the scalable MDP formalism was a bit hard to understand. Towards this end, we simplified the Figure 2 and provided how we go about scaling the MDP. We also simplied the caption by explaining everything in plain english.

---

> > ### Author Response · Authors · 2022-08-02
> > **Author Response Contd..**
> >
> > **Intuition of Scaling Function:** We really like your idea of starting with a simple example to demonstrate key ideas at the beginning of our experiments section. Towards this end, we have conducted experiments involving a simple two layer neural network learning with episodic reinforce in a 3d 10x10x10 grid world environment. We have also added additional details in Example 1 of Section 6 to make clearer connections with the formalism of of scalable MDPs (e.g. explicitly stating the form of the scaling functions). Moreover, as highlighted in the figure provided in our comment titled "Motivating Experiments", our experiments in this domain demonstrate how at a fixed number of training steps per task, final performance is negatively effected by increasing the time between task switches $\tau$ and the number of tasks $|\mathcal{Z}|$. We believe this experiment further highlights the effects of myopic bias and is nice in that it allows us to generate 1,000 different and equally difficult tasks due to its synthetic nature.
> >
> > **Myopic Bias of Value-based Approaches (FQI):** Value-based approaches struggle with myopic bias due to the staleness of policy evaluation during bootstrapping-style evaluation as described in the paragraph leading up to Corollary 3 in Section 6. With respect to fitted-Q-iteration specifically, we believe this is can be seen in its statement in equation 2 of (Chen and Jiang, 2019) where $f_{k-1}$ is used to produce $f_{k}$ with Bellman style lookahead updates where bootstrapping is based on the previous estimate for all steps greater than one ahead. As such, this procedure must be run a number of times reflecting the mixing time for each policy to avoid staleness bias in the policy improvement steps. We do not fully follow your train of thought related to coverage, but presume this is related to the kind of coverage that is needed to demonstrate the convergence of value-based approaches like Q-learning. Approaches may indeed eventually converge to the optimal policy while only taking one-step lookahead updates, but the individual updates are still myopic, which limits sample efficiency in environments with high mixing times as highlighted in Corollary 4. Please let us know if we did not correctly understand your question and we would be more than happy to clarify further.
> >
> > **Connection Between Scalable MDPs and Continual Atari:** Thank you for highlighting that this was an area of confusion. We have added a description to further clarify the connection with the terminology of Section 3 in Example 1 of Section 6. Please let us know if there are any additional details that it would be helpful to provide in this section as well.

---

> > > ### Author Response · Authors · 2022-08-08
> > > **Have the revisions addressed your concerns?**
> > >
> > > Thank you again for your thoughtful review. In our rebuttal, we have addressed your concerns regarding
> > > 1) connections to the literature on continual RL
> > > 2) scalable MDPs intuition and its connection to continual Atari
> > > 3) simple experiments to build intuition (please refer to this plot [https://i.imgur.com/YtYZFlC.png](https://i.imgur.com/YtYZFlC.png)).
> > >
> > > Since the discussion phase is closing soon, we would like to know if you have any remaining concerns that we haven't addressed. We will be more than happy to clarify.
> > >
> > > Thanks

---

### Official Review · Reviewer_hA9x · 2022-07-11

**Rating:** 5
**Confidence:** 2
**Soundness:** 3 good
**Presentation:** 3 good
**Contribution:** 3 good

**Summary:**

In this paper, the authors propose scalable MDPs in which the MDPs can be defined using the scale parameter. Using scalable MDPs, the learned algorithm can be evaluated on small scale problem to large scale problem. Based on the notion of mixing times, they show that the scalable MDPs have mixing times that scales polynomially. In the experiments, they show the empirical analysis of mixing behavior on Atari games, and the results show that the mixing time on given scalable MDPs is a function of relative error and the number of tasks.

**Questions:**

Specified in Cons.

**Strengths And Weaknesses:**

Pros:

The definitions of each component on scalable MDPs are well explained, and the theorems and experimental results are consistent.

Cons:

1. It would be better to carry out the experiments on Roboschool or MuJoCo that have vector-valued state space. Is the mixing time much smaller than the original experiment?

2. Though it focuses on the scalable MDPs in terms of the continual reinforcement learning, the performance of continual learner in this setting is not mentioned. As a result, it was hard to notice the importance of the mixing time in scalable MDPs for the continual reinforcement learning scenario

---

> ### Author Response · Authors · 2022-08-02
> **Author Response**
>
> Thank you for your review and kind words about the theoretical results and experiments provided in our submission.
>
> **Vector-valued state space:** We really appreciate your suggestion regarding vector-valued state spaces and the potential for them to induce smaller mixing times. Inspired by your suggestion, we have provided additional experiments across 5 Mujoco environments, `HalfCheetah, Hopper, Walker2d, Ant, Swimmer`,  in Figure 5 and Section 6.1. We are presenting the plot again here just for quick reference (here different colors correspond to different algorithms: SAC, A2C, and PPO in order): [https://imgur.com/a/dPGUli5](https://imgur.com/a/dPGUli5.png)
>
> It turns out that the mixing times are overall very similar to those experienced in the analogous Atari settings. This is consistent with our expectation based on Example 1 as the mixing time is fundamentally driven by the scaling of $\tau$ and $|\mathcal{Z}|$ in both cases.
>
>
> **Performance of Continual Learners:** In the original submission our paragraph at the beginning of Section 7 titled "Optimization Instability for Continual Atari" highlighted a number of existing papers that have learned on similar continual Atari settings and noted significant optimization instability. This was part of our rationale for exploring Atari to begin with and we believe that highlighting this consensus in the literature is stronger than only showing this for a limited set of experiments and baselines. However, we do appreciate your comment and feel that it could be complementary to a comment made by reviewers oAr4 and VEPZ about how it would be nice to start with a simple synthetic experiment to motivate the experiments on larger domains that come later. Towards this end, we have conducted experiments involving a simple two-layer neural network learning with episodic reinforce in a 3d 10x10x10 grid world environment. As highlighted in the figure in our comment titled "Motivating Experiments", our experiments demonstrate how at a fixed number of training steps per task, final performance (after continual learning is complete) is negatively affected by increasing the time between task switches $\tau$ and the number of tasks $|\mathcal{Z}|$. We believe this experiment further highlights the effects of myopic bias during learning and is nice in that it allows us to generate 1,000 different equally difficult tasks due to its synthetic nature.

---

> > ### Author Response · Authors · 2022-08-08
> > **Have the revisions addressed your concerns?**
> >
> > Thank you again for your review. In our rebuttal, we have addressed your concerns regarding
> >
> > 1) more experiments leveraging the Mujoco environment (please refer to  Figure 5 and Section 6.1 or this link [https://imgur.com/a/dPGUli5.png](https://imgur.com/a/dPGUli5.png))
> > 2) a discussion on how mixing times affect learning by providing results on a simple task (please refer to this plot [https://i.imgur.com/YtYZFlC.png](https://i.imgur.com/YtYZFlC.png)).
> >
> > Since the discussion phase is closing soon, we would like to know if you have any remaining concerns that we haven't addressed. We will be more than happy to clarify.
> >
> > Thanks

---

### Official Review · Reviewer_oAr4 · 2022-07-11

**Rating:** 7
**Confidence:** 3
**Soundness:** 3 good
**Presentation:** 3 good
**Contribution:** 3 good

**Summary:**

This paper investigates the effect of scalability and mixing time in MDPS for RL. The Authors introduce scalable MDPs as a family where the state space or a region of that can be scaled subject to a scaling parameter and a scaling operation. They show that any scalable MDP has a polynomial mixing time with respect to state space. The authors also discuss that traditional RL algorithms (monte carlo and bootstrapping) cannot scale well  to larger size problems because of the myopic bias which slows down learning. They conclude the paper by showing that polynomial mixing time can emerge in continual RL tasks (atari environments) and analyze instability caused by this bias.

**Questions:**

In empirical studies the average mixing time is used instead of the actual mixing time (with maximum), would you explain how this might change your settings and empirical results? for example if we change the starting state distribution, how does your empirical result change?

This paper is on the analysis side of the problem that arises because of the polynomial mixing time, I was wondering if authors can provide some clues on possible approaches to solve the issue?

**Limitations:**

In section 7, Authors have discussed the limitations of their paper from different perspective.

**Strengths And Weaknesses:**

Strengths: Paper is fairly well-written, the topic of this paper and analyzing mixing time induced by policies is fundamental to RL. The flow of the paper on theorems and corollaries and the discussion after was enjoyable to read.
Weakness: For the purpose of reach in the RL community,  I think it would be nice if the authors could start with a simpler markov chain in the empirical section to show the mixing time, bias and etc, before jumping to atari environments

---

> ### Author Response · Authors · 2022-08-02
> **Author Response**
>
> Thank you for your kind words about the writing and contribution of our paper as well as your thoughtful review.
>
> **Simple Experiments:** We really like your idea of starting with a simple example to demonstrate key ideas at the beginning of our experiments section. Towards this end, we have conducted experiments involving a simple two-layer neural network learning with episodic REINFORCE in a 3-dimensional 10x10x10 grid world environment. As highlighted in the figure we have uploaded in our comment titled "Motivating Experiments", our experiments demonstrate how at a fixed number of training steps per task, final performance is negatively affected by increasing the time between task switches $\tau$ and the number of tasks $|\mathcal{Z}|$. We believe this experiment further highlights the effects of myopic bias and is nice in that it allows us to generate 1,000 different equally difficult tasks due to its synthetic nature.
>
> **We are pasting the link here for easy accessibility: [https://i.imgur.com/YtYZFlC.png](https://i.imgur.com/YtYZFlC.png)**
>
> **Maximum vs. Average Mixing Times:** Thank you also for your important question about taking the average over start states as opposed to the maximum in our mixing time calculations. Our motivation in this regard is that the initial $\epsilon$-return mixing time from (Kearns and Singh, 2002) was defined as a worst-case metric so that it can be leveraged to establish formal regret bounds. We were fearful that if we just reported the standard $\epsilon$-return mixing time we could be criticized for reporting inflated values beyond those that agents actually encounter in practice. We have addressed this in two ways:
> 1) We have normalized $\epsilon$ in terms of a relative error metric so that we can ensure $\epsilon$ is significant relative to the true reward rate.
> 2) We have reported the mean over start states as opposed to the worst case to reflect typical performance over arbitrary on-policy initial states.
>
> Moreover, it is computationally intractable to compute the mixing time from all possible start-states at once, as reservoir sampling may skew this distribution in a way it does not for the mean. With the additional space in the final revision, we will also add results for the maximum over the start-states in the reservoir. Our finding is that the general pattern remains the same as a function of $\tau$ and $|\mathcal{Z}|$, but that the mixing time numbers are even higher.
>
> **Clues About Possible Approaches:** This is of course a very important direction that we hope future work will be able to address now that our paper has established the problem. In the submitted draft, we tried to partially speak to this in the paragraph on “Development of New Approaches” in Section 7. We have reorganized this section based on your comment which is now called "Inspiring New Approaches for Continual RL". This has allowed us to further highlight the discussion from our submitted draft about approaches for directly estimating the steady-state distribution. Additionally, we have added discussion based on your comment to also highlight tracking approaches [1] that move beyond the idea of finding converging stationary policies in RL as a potential solution.
>
> [1] Richard S Sutton, Anna Koop, and David Silver. On the role of tracking in stationary environments. In Proceedings of the 24th international conference on Machine learning, pages 871–878, 2007.

---

> > ### Author Response · Authors · 2022-08-08
> > **Have the revisions addressed your concerns?**
> >
> > Thank you again for your thoughtful review. In our rebuttal, we have addressed your suggestions regarding
> >
> > 1) simple experiments (please refer to this plot [https://i.imgur.com/YtYZFlC.png](https://i.imgur.com/YtYZFlC.png))
> > 2) confusion between average vs. maximum mixing times
> > 3) a discussion of possible approaches to address polynomial mixing.
> >
> > Since the discussion phase is closing soon, we would like to know if you have any remaining concerns that we haven't addressed. We will be more than happy to clarify.
> >
> > Thanks

---

### Author Response · Authors · 2022-08-02
**Motivating Experiments**

With the additional space allotted for the final draft, we plan on adding motivating experiments that enable us to address concerns raised by each reviewer prior to Section 6.1 of the revision.

**The link to the plot can be found here: [https://i.imgur.com/YtYZFlC.png](https://i.imgur.com/YtYZFlC.png)**
### 3D Grid World Domain

**Setup:** We consider a simple 10x10x10 3-dimensional grid world environment with 6 actions corresponding to up and down actions in each dimension. In this environment, we can define 1,000 different tasks corresponding to considering each unique state as a goal location.

**Inputs:** At each time-step the agent is sent a 60-dimensional concatenation of 6 10-dimensional one-hot vectors. The first three one-hot vectors correspond to the agent's x,y, and z coordinates and the second three correspond to the goal location.


**Connection to Continual RL:** As we have motivated in Section 2.2, this environment is made to demonstrate the difficulty associated with myopic optimization bias even when task boundaries and their relation are fully observable. We consider a continual episodic RL formulation following the description in Example 1 where task sequences arise from changing the goal location every $\tau$ steps (the set of goal locations defines the set of tasks, $\mathcal{Z}$). As such, the diameter of the MDP clearly increases as $\tau$ and the number of tasks $|\mathcal{Z}|$ are scaled up, leading to at least linear scaling. As a result, an agent that updates only based on the current episode experiences myopic bias with respect to the steady-state distribution (over which the tasks are balanced).

**Training Procedure:** The 60-dimensional sparse observation vector is processed by a multi-layer perceptron with two 100 unit hidden layers and Relu activations. The agent performs optimization following episodic REINFORCE. We explore different configurations of this scalable MDP by varying $|\mathcal{Z}|$ from 1 to 1,000 and varying $\tau$ from 100 to 10,000. An important baseline to consider is the extreme of switching tasks after every episode as in multi-task RL (which is generally considered an upper bound on continual RL performance). The agent receives a reward of 1 every step it makes progress towards its goal and a reward of 0 otherwise. This implies that the reward rate of $\pi^*$ is 1.0 and that the reward rate of a uniform stochastic policy over actions is 0.5 for every task regardless of the length and goal location.

**Empirical Findings:** In the Figure above, for each $\tau$ and $|\mathcal{Z}|$, we train the agent for 10,000$|\mathcal{Z}|$ total steps, switching between tasks every $\tau$ steps. We report the average reward rate at the end of training across the randomly selected task progressions as an average across 60 seeds. It is interesting to notice the episodic transition performance, which can near flawlessly learn a single task in 10,000 steps but struggles significantly with interference in the multitask setting. Performance goes down from 1 to 30 tasks and appears to rebound afterward, which is logical because the similarity between incoming tasks and old tasks is increasing and the total number of training steps across tasks is going up. When both $\tau$ and $|\mathcal{Z}|$ are high at the same time, learning performance is quite poor (not much better than a uniform policy). The starting location is always in the same corner, so some commonalities can be exploited even by a policy experiencing significant interference. Performance bottoms out with fewer tasks added when $\tau$ is larger, which makes sense as the myopic bias of optimization is greatest in that setting.

---

### Meta-Review · Area_Chair_wp6k · 2022-08-24

**Recommendation:** Accept
**Confidence:** Certain

**Metareview:**

This paper investigates the effect of scalability and mixing time in MDPS for RL. The Authors introduce scalable MDPs as a family where the state space or a region of that can be scaled subject to a scaling parameter and a scaling operation. They show that any scalable MDP has a polynomial mixing time with respect to state space. The authors also discuss that traditional RL algorithms (monte carlo and bootstrapping) cannot scale well to larger size problems because of the myopic bias which slows down learning. They conclude the paper by showing that polynomial mixing time can emerge in continual RL tasks (atari environments) and analyze instability caused by this bias.

This is a well written where each component on scalable MDPs are well explained, and the theorems and experimental results are consistent.  All four reviewers were on the positive side for acceptance.



**Award:**

No

---

### Decision · Program_Chairs · 2022-09-14

Accept